Clim. Past Discuss., doi:10.5194/cp-2015-170, 2016 Manuscript under review for journal Clim. Past Published: 15 January 2016 © Author(s) 2016. CC-BY 3.0 License.

This discussion paper is/has been under review for the journal Climate of the Past (CP). Please refer to the corresponding final paper in CP if available.

# Dynamical downscaling of the western North Pacific from CCSM4 simulations during the last glacial maximum and late 20th century using the WRF model: model configuration and validation

# J. Yoo and J. Galewsky

University of New Mexico, 221 Yale Blvd. NE MSC03 2040, Albuquerque, NM 87131-0001, USA

Received: 11 November 2015 - Accepted: 9 December 2015 - Published: 15 January 2016

Correspondence to: J. Yoo (jinwoong.yoo@gmail.com)

Published by Copernicus Publications on behalf of the European Geosciences Union.

## Abstract

Using the Weather Research and Forecasting (WRF) model (version 3.5.1), dynamical downscaling of the Community Climate System Model, version 4 (CCSM4), simulations of the last glacial maximum (LGM) and 20th century (ensemble member #6) run

- <sup>5</sup> were conducted to simulate ten years of climate over the western North Pacific during the LGM and modern climates, respectively. This paper describes the downscaling procedures for the Weather Research and Forecasting (WRF) model experiments and the quantitative and qualitative model validations comparing with the CCSM4 LGM and 20th century simulations results.
- Results of the dynamical downscaling of the CCSM4 LGM paleoclimate and twentieth century using the WRF model show not only that the WRF model is capable of long-term simulations in the paleoclimate state of LGM, but also that the WRF model can correct biases in the general circulation model (GCM), producing more realistic spatial distributions of the pressure-level variables. The downscaling of a GCM model
- <sup>15</sup> using the WRF model (36 km) for the regional climate simulation is considered computationally cost-effective and reliable from the perspectives of model thermodynamics in general, although there are some model errors still existing with dynamic variables.

## 1 Introduction

Over the past 21 000 years, the Earth has undergone a substantial warming induced
 <sup>20</sup> by natural vacillations in orbital geometry, a concomitant rebound in greenhouse gas levels, and changing boundary conditions (i.e., ice sheet retreat and rising sea level). At the height of the last glacial maximum (LGM) 21 000 years ago (21 ka), the drop in CO<sub>2</sub> levels to 185 ppm, the drop in CH<sub>4</sub> to 350 ppb, and the far greater extent of ice coverage at high latitudes, are the most important forcing changes for the climate of
 <sup>25</sup> the LGM, while seasonal and latitudinal distribution of incoming solar radiation at the top of Earth's atmosphere was the second largest difference from those of today (e.g.

Otto-Bliesner et al., 2006). Thus, surface temperatures in the LGM were about 2°C lower in the tropics (Broccoli, 2000) and about 30°C colder over the Laurentide ice sheet (Braconnot et al., 2007).

- Downscaling of the paleoclimate can provide insights about the paleoclimate conditions that cannot be obtained otherwise by just compiling the proxy records. For example, horizontal and vertical spatial distributions of variables of interest can be inferred or conjectured realistically through the downscale modeling considering the large-scale climate condition as well as the proxy information. Geological studies have speculated about what synoptic scale patterns might have changed in the tropics, but global model simulations of paleoclimates offer synthetic data to compare with results from geologic proxies (Galewsky et al., 2006). Therefore, downscaling of general circulation model
- (GCM) output can provide a quantitative foundation for paleoenvironment research in a variety of applications.
- The goal of this study is to downscale of the Community Climate System Model (ver-<sup>15</sup> sion 4; CCSM4) LGM paleoclimate and twentieth century runs from the phase five of coupled model intercomparison project (CMIP5) and paleoclimate model intercomparison project version 3 (PMIP3) to understand the behavior of large-scale dynamics and thermodynamics over the western North Pacific under the LGM and present eras using the Weather Research and Forecasting (WRF) model. Specifically, the purpose of this paper is to address the following: (1) procedures to conduct a dynamical downscaling
- of the CCSM4 model outputs for the LGM and late modern simulations; and (2) evaluation of the downscaling performance of the WRF model by comparing the downscaling results with the GCM LGM paleoclimate and twentieth century simulation results.

This paper is organized as follows. In Sect. 2, we describe the preparation methods
 and procedures for the WRF model simulation set-up, specifically, for the LGM period.
 We discuss the validation of downscaling experiment results in Sect. 3. A discussion will follow in Sect. 4. A summary and conclusion are addressed in Sect. 5.