# Peer review of "Dynamical downscaling of the western North Pacific from CCSM4 simulations during the last glacial maximum and late 20th century using the WRF model: model configuration and validation"

_Climate of the Past, 2015_

## Referee Comment (RC1) · Anonymous Referee #1 · 22 Jan 2016

General comments

This paper deals with dynamical downscaling of the coupled atmosphere-ocean general circulation model CCSM4 outputs with WRF, on the western North Pacific, both for modern and Last Glacial Maximum climates. I am sorry to say that I have 3 general comments which, to my opinion, strongly prevent the publication of this paper in Climate of the Past.
[Figure]

First, the modelling work is, to my opinion, questionable because the authors integrate only on 10 years, which is much too small to be representative of a climatological mean (typically at least 30 to 50 yrs for atmospheric variables, and 50-100 yrs for oceanic variables). Additionally, for the LGM, they use CCSM4 outputs which are not at equilibrium for the LGM as boundary conditions for WRF (see specific comment below).

The second comment is the lack of a clear scientific question. What is the scientific question underlying your work ? It is not explained. The authors state page 15 that : "the goal of this study is to investigate the behavior of large-scale dynamic and thermo-dynamic variables in the downscaling experiments over the western North Pacific under the LGM and modern climates". This is not a scientific question, rather it is a mean to achieve something, but we don't know what they want to achieve. For example, why do you choose this area in particular (and not, let's say, the eastern tropical Pacific) ? Why do you choose Last Glacial Maximum and modern (and not, let's say, mid-Holocene)? I am sure you have good reasons to do this, but they are not mentioned. You should start by clarifying the scientific question.

The third important comment is that you cannot say that you validate your model if you do not confront it to observations and paleo-reconstructions. You cannot state in the abstract that "the WRF model corrects biases of the GCM, producing more real-istic spatial distributions of the pressure-level variables" if you do not compare your WRF outputs to observations (for 20th century simulation) and paleo-reconstructions of temperature, precipitation and SSTs for the LGM simulation. Therefore, the goal of your paper, which is to validate the WRF model on the western North Pacific, is, to my opinion, missed. Calculating the root mean square error (RMSE) between the WRF and the CCSM4 simulation can only tell you how much these two simulations differ between them, but not how much they both differ from 'reality'. In other words, the CCSM4 simulation can certainly not be considered as an observation.

Consequently, the authors need to 1/ pose a clear scientific question for which the

downscaling is a tool used to answer the question, not a goal in itself ;2/ carry out a proper model-data comparison both for 20th century and LGM on their area of interest, in order to validate their modeling exercise and 3/ use CCSM4 outputs for the LGM coming from the end of the simulation (after 400 or 500 years of spin-up), not from the beginning (years 1 to 11 here used) and integrate on 50 to 100 years.

As the authors state, for LGM there is not much data on the global scale. However, there is a cluster of pollen data right in their area of interest on land (see synthesis in Bartlein et al., 2011). For oceanic data, there are also a few SST reconstructions at the southern fringe of their area of interest (see MARGO, 2009). It will be worth to check in the literature if new SST reconstructions have been published in this area since 2009 (I have no idea about this unfortunately). I know it is not easy to go through paleo-reconstructions when you are a modeller. But they could at least have done the model-data comparison for the 20th century simulation, comparing with observations. For the LGM, I would suggest that the authors use the Bartlein et al., 2011 database for land, and the MARGO 2009 synthesis for the ocean + potential new SST reconstructions (see a model-data comparison in Kageyama et al., 2013). If the authors are concerned by the difficulties of multi-proxy databases (see for example Leduc et al., 2010 for the difficulties in the interpretation of oceanic proxies), they can choose to do a comparison only with one type of proxy (for example, only with Mg/Ca on the ocean ; or comparing separately with the Mg/Ca and the alkenone data available in MARGO). But in any case, a model-data comparison is absolutely needed for validation of the model. If the model-data comparison is better with the small-scale variables calculated with WRF than with the large-scale variables of CCSM4, then that means you can validate WRF.

Specific comments Page, 2, line 12 : "long-term simulations". This term is not appropriate since these simulations are equilibrium climate, not transient. Rather use "paleoclimate simulations".

Page 3, line 8 "variables can be inferred . . . through the downscaled modelling considering the large-scale climate conditions as well as the proxy information". To me,

this sentence suggests that you included data assimilation into your downscaling pro-
cedure, which is not the case. Please modify this sentence for more clarity.

Page 5, 2.3 retrieving CCSM4 data Please explain clearly what you mean by "20th
century simulation" and what is the corresponding reference paper for this simulation.
Is it the simulation of the historical period ? It is equilibrium or transient ?

Page 7, lines 15-20 "Since 1870 was the initialization year for the LGM simulation,
we chose year 1871 to be the first year of the LGM simulation to avoid any issue
associated with the CCSM4 model spin-up" This sentence shows to my opinion that the
authors are not used to dealing with equilibrium paleoclimate simulation. In particular
for the LGM, for which boundary conditions are very different from the modern, the
climate model generally takes several centuries to get the ocean at equilibrium with
these different boundary conditions. Typically for the LGM, this would take at least 400
years. Therefore, the authors would need to take the CCSM4 outputs after 400 years
of simulation. The best would be to take the last years of the simulation, where it is
sure that the model has achieved equilibrium.

Page 7 'validation of model results' The calculation of RMSE between CCSM4 and
WRF is not a proper way to validate the use of WRF. Validation should be done through
a comparison of CCSM4 and WRF 20th century simulations with observations ; and
through a comparison of CCSM4 and WRF LGM simulations with paleo-temperature
and paleo-precipitation reconstructions (see for example Kageyama et al., 2013 for a
model-data comparison for the LGM on a global scale).

I will not comment the rest of the analysis since it is based on the RMSE between
CCSM4 and WRF, which I consider irrelevant for the validation of WRF.

References : Bartlein PJ, Harrison SP, Brewer S, Connor S, Davis BAS, Gajewski K,
Guiot J, Harrison-Prentice TI, Henderson A, Peyron O, Prentice IC, Scholze M, Seppa
H, Shuman B, Sugita S, Thompson RS, Viau AE, Williams J, Wu H (2011) Pollen-
based continental climate reconstructions at 6 and 21 ka: a global synthesis. Clim

Dyn, 37:775–802

Kageyama M et al. (2013) Mid-Holocene and last glacial maximum climate simulations with the IPSL model: part II: model-data comparisons, Clim Dyn, 40:2469–2495

Leduc, G., R. R. Schneider, J.-H. Kim, and G. Lohmann (2010), Holocene and Eemian sea surface temperature trends as revealed by alkenone and Mg/Ca paleothermometry, Quat. Sci. Rev., 29, 989–1004.

MARGO project members: (2009) Constraints on the magnitude and patterns of ocean cooling at the last glacial maximum. Nat Geosci 2:127–132

---

## Referee Comment (RC2) · Anonymous Referee #2 · 28 Apr 2016

General comment: The authors performed a dynamical downscaling experiment with a regional climate model (WRF) based on GCM (CCSM4) output as boundary conditions for glacial (LGM) and present day climate conditions, respectively. Unfortunately (I agree with Referee #1), this study exhibits some clear deficits, which prevents this manuscript for being published in Climate of the Past.

Major Points: (1) A general problem of this study is the missing scientific question. This study only compares two different model data sets (which are not independent since

[Figure]

GCM data is used as boundary conditions for downscaling) without explaining why this is of importance.

(2) In general, models can only be validated by comparing model output with observational data and not by comparing them solely against each other. This study completely ignores both (i) observational data (e.g., reanalysis datasets ERA, NCEP,. . . ) for evaluation of present day performance of the models and (ii) paleoclimate proxy data to validate the ability of the models to simulate the climate under glacial conditions (refer to proxy data sources from referee #1). How do you know that the WRF model corrects biases of the GCM if you don't validate the model against observations?

(3) Although the dynamical downscaling seems to be technically correct (considering glacial boundary conditions e.g., ice sheets, drop of sea level), the length of the simulations of only 10 years seems to be bit to short. I wouldn't say that 50 or even 100 years are needed for comparison of the different time slices (since this is implies high computational costs), but in my opinion you should simulated at least a period of 20 years to create a sufficient amount of data. In contrast to referee #1, I think the CCSM4 LGM run is in equilibrium state (if you used the data as stored in the PMIP3/CMIP5 database), since the PMIP3 data are only the last 100 years of the corresponding model run (model spin-up has been considered for all PMIP3 LGM experiments while only data since equilibrium was reached is stored in the PMIP3/CMIP5 database). A question arises which run you precisely take for present day conditions? Is it the historical run, which is NOT in equilibrium since it is forced by changing greenhouse gas concentrations, or do you use the output of the pre-Industrial run, which uses constant greenhouse gas (GHG) concentrations during the whole integration? Would it make sense to compare an equilibrium run to a model run with additional internal forcing? This is not mentioned in the manuscript.

Minor comments and examples for major comments (1) - (3)

P2L8: Model validation requires observational data

P2L17: How do you know about model errors when you don't compare to observational data?

P2L19: Is the warming induced by ice sheet retreat and rising sea level (this is how it reads), or is this a consequence of warming?

P2L22: Is the drop of GHG concentrations and the greater extend of ice sheets really the important forcing changes or are they just a consequence of reduced incoming solar radiation? How can you precisely rate theses different forcing effects and consequences?

P3L15: Reference missing for CCSM4

P3L14: should read "...to downscale the Community..."

P3L26: What is discussed in Sect.4 if you discuss already in Sect. 3? Be more precise.

P4L4: Version 3.5.1?

P4: Citations for all considered parameterisations is missing.

P5L2: Why PMIP2? Shouldn't you use recommendations for PMIP3? Or are they the same?

P5L17: "ice sheets over" rather then "ice sheets in"

P6L2: What is tropical channel?

P6L24: What is MOAR? Which simulations do you exactly use for present day? Is it also a PMIP3/CMIP5 run? Needs clarification.

P7L17: Year 1871 is definitely not the second year of the LGM simulation. PMIP3-runs are all performed with a sufficient spin-up time until the model reaches equilibrium. The PMIP3 output in the database contains only the last 100 years of the simulation.

P7L19: Why do you reinitialize your model each year? This ignores internal model dynamics at the start of each year. If you consider only winter (DJF) for analysis, you

might get different results. Since you used 6-hourly data as forcing data you don't have to reinitialize the model each year (unless there is a reason you didn't mention. . .)

Chapter3 and rest: Validation of model results makes only sense when considering observational/proxy data. Since this is totally missing, I'm not going into details.

P16L28: You should acknowledge for the PMIP3/CMIP5 data as suggested on their homepage.

---

## Author Comment (AC1) · 25 May 2016

Authors thank the anonymous reviewers for their kindness with valuable comments and suggestions. We, here, are trying to provide answers to the comments in a unified fashion.

Authors ran the dynamical downscaling simulations to understand the differences and/or similarities in large-scale atmospheric dynamics and thermodynamics between

the LGM paleo-environment and the present over the eastern North Pacific domain, from which we are expecting that many interesting research questions can develop. In fact, comparisons between the two geologic time periods can provide many scientific insights to understand the unfolding climate changes in many perspectives including extreme climate events. Authors agree with the reviewers that including these statements would help readers understand the manuscript more clearly.

The CCSM4 LGM simulations are at equilibrium with the prescribed LGM boundary conditions as per PMIP3 protocols. The CCSM4 LGM simulation output that was used in this dynamical downscaling experiment was part of the IPCC-TIER1 CCSM4 Last Glacial Maximum. It is also part of the PMIP3/CMIP5 and is archived with its case name of b40.lgm21ka.1deg.003. The entire CCSM4 LGM simulation was run for one thousand years (Brady et al. 2013). Brady et al. (2013) analyzed the last 30 years of the 1000-year long LGM simulation. Authors used the first 10 years of the last 30 years of the CCSM4 LGM simulation. Please refer to Brady et al. (2013) for further detail of the CCSM4 LGM simulations. 1870 reference year was the year the NCAR scientist started to reproduce the CCSM4 LGM simulation output at the higher temporal resolution of six hour interval. There should not be any issue of equilibrium even with the 1870 reference year for the restart of the simulation but authors wanted to minimize any potential issues if possible. Although ten-year long integration of the dynamical downscaling may not be long enough to fully appreciate the paleoclimate in detail, authors think that even ten-year simulation can suggest a close-to-general condition of the large-scale atmospheric environments at the time of the LGM over the eastern North Pacific domain due to the use of the CCSM4 LGM simulation at equilibrium.

To our best knowledge, there is no paper published for the CMIP5 CCSM4 20th century simulation. However, Brady et al. (2013) can be referred to for further information about the CCSM4 20th century simulation as well. Also, detail description for the CCSM4 1 degree 20th Century Ensemble Member #6 (MOAR) data (Case Name: b40.20th.track1.1deg.012) can be found at www.cesm.ucar.edu/experiments/cesm1.0.

Specifically, the CCSM4 20th century MOAR simulation was run for 1300 years (Brady et al. 2013). The NCAR re-produced 6-hourly outputs for the 20th century (1950-2005) from the 1300 years simulation. Please note that the CCSM4 20th century simulations are NOT at equilibrium because they have been forced with transient greenhouse gases and aerosols (among other things). Authors acknowledge that there is a limitation in the comparison between the 20th century simulation and the LGM simulation because the clearest comparison with the LGM simulations would be with a preindustrial control, which is a true equilibrium climate. Although, we wanted to compare the LGM paleo-environment against the modern condition rather than the preindustrial condition, which would give us more intuitive comparisons with the large-scale atmospheric dynamic and thermodynamic features over the eastern North Pacific.

Regarding the downscaling experiments and the purpose of the manuscript, there are a few things that we would like to make clear. There is no data assimilation applied to our downscaling simulations. It should be understood also that this manuscript is to assess the dynamical downscaling performance of the WRF model as a regional climate model, not to validate the CCSM4 LGM simulations against paleo-environmental proxy records. For the validation of the CCSM4 LGM simulations, Brady et al. (2013) compared the CCSM4 LGM simulation results of mean annual LGM surface temperature with the Multi-proxy Approach for the Reconstruction of the Glacial Ocean Surface (MARGO) reconstruction (Waelbroeck et al. 2009) and mean annual temperature (MAT) differences from the land-based reconstruction of Bartlein et al. (2011). Additional land-based proxy data from Schmittner et al. (2011), available at http://mgg.coas.oregonstate.edu/;andreas/ data/schmittner11sci/ and those not included in Bartlein et al. (2011), are also utilized for the model validation in Brady et al. (2013), including estimates from ice cores over Greenland and Antarctica. However, since the validation of the CCSM4 LGM simulations has been provided in Brady et al. (2013), comparisons between the GCM (CCSM4 LGM) simulations and the regional downscaling simulations are necessary to evaluate the dynamical downscaling performance of the WRF model as a regional climate model, which is the main purpose

of this manuscript. Please refer to Brady et al. (2013) for the comparisons between the CCSM4 LGM simulation and various proxy records. Nevertheless, authors agree that it will improve our model validation manuscript to compare the downscaling results with the latest reconstruction datasets of paleoenvironment for the LGM such as the MARGO reconstruction data as far as the proxy data exists and observation data for the modern period. Model-to-proxy data and model-to-observation data comparisons would provide valuable information whether downscaling of the CCSM4 using the WRF model performs better than the GCM or not both in the LGM and modern periods.

Reference Brady, E. C., B. L. Otto-Bliesner, J. E. Kay, And N. Rosenbloom, 2013: Sensitivity to Glacial Forcing in the CCSM4, Journal of Climate, 26, 1901-1925.